# Effect of Different Light Wavelengths on *Zymoseptoria tritici* Development and Leaf Colonization in Bread Wheat

**DOI:** 10.3390/jof9060670

**Published:** 2023-06-14

**Authors:** Minely Cerón-Bustamante, Francesco Tini, Giovanni Beccari, Paolo Benincasa, Lorenzo Covarelli

**Affiliations:** Department of Agricultural, Food and Environmental Sciences, University of Perugia, Borgo XX Giugno 74, 06121 Perugia, Italy; minely.ceronbustamante@studenti.unipg.it (M.C.-B.); francesco.tini@unipg.it (F.T.); paolo.benincasa@unipg.it (P.B.); lorenzo.covarelli@unipg.it (L.C.)

**Keywords:** fungal pathogens, light, wheat, *Zymoseptoria tritici*, Septoria tritici blotch, red wavelength, blue wavelength

## Abstract

The wheat pathogen *Zymoseptoria tritici* can respond to light by modulating its gene expression. Because several virulence-related genes are differentially expressed in response to light, different wavelengths could have a crucial role in the *Z. tritici*–wheat interaction. To explore this opportunity, the aim of this study was to analyze the effect of blue (470 nm), red (627 nm), blue–red, and white light on the in vitro and in planta development of *Z. tritici*. The morphology (mycelium appearance, color) and phenotypic (mycelium growth) characteristics of a *Z. tritici* strain were evaluated after 14 days under the different light conditions in two independent experiments. In addition, bread wheat plants were artificially inoculated with *Z. tritici* and grown for 35 days under the same light treatments. The disease incidence, severity, and fungal DNA were analyzed in a single experiment. Statistical differences were determined by using an ANOVA. The obtained results showed that the different light wavelengths induced specific morphological changes in mycelial growth. The blue light significantly reduced colony growth, while the dark and red light favored fungal development (*p* < 0.05). The light quality also influenced host colonization, whereby the white and red light had stimulating and repressing effects, respectively (*p* < 0.05). This precursory study demonstrated the influence of light on *Z. tritici* colonization in bread wheat.

## 1. Introduction

Light is a crucial abiotic element that modulates several physiological processes in living organisms. Although light contains a wide range of wavelengths, only a small part is both detectable and useful for lifeforms [1]. Particularly, plants can recognize ultraviolet radiation (260–380 nm), the visible spectrum of light (380–740 nm), and far-red radiation (700–850 nm) via specific photoreceptor proteins that are adept at sensing the quality, quantity, and duration of wavelengths [2,3,4,5]. Nevertheless, light sensing is not an exclusive propriety of autotroph organisms. For instance, fungi also have several plant homolog photoreceptors that are used to sense light [6,7,8,9]. In fungi, photoreceptors perform signaling functions during growth, reproduction, and secondary metabolism [6,7,10].

Light also plays an important role in fungal pathogenicity and virulence by stimulating spore germination, tube germ tropism, penetration, and host colonization [11,12,13]. For instance, in the maize pathogen *Cercospora zeae-maydis*, light is required for stomatal tropism and appressorium formation [11]. Similarly, in the gray mold causal agent *Botrytis cinerea,* near-ultraviolet (UV) radiation and blue light promoted infection-hypha formation on the broad bean leaf surface [13]. Moreover, light modulates the synthesis of secondary metabolites such as toxins and pigments, which are important compounds for infections in some fungi. For example, light increases virulence in *Colletotrichum acutatum* by inducing the biosynthesis of melanin, a crucial compound for host colonization [12]. Additionally, blue light is essential for the synthesis of the phytotoxin cercosporin in *C. zeae-maydis* [11].

Light can also have a detrimental effect on plant–pathogen interactions by acting either on the infection process or fungal fitness [14,15,16,17]. For example, light reduced infection by both *Magnaporthe oryzae* and *Monilinia laxa* in artificially inoculated rice plants and nectarine fruits, respectively [18,19]. Similar results have also been observed in *B. cinerea* infecting *Arabidopsis thaliana*. In *A. thaliana*, light diminished gray mold lesions in comparison with plants grown in the dark [20]. It has been suggested that the reduction in grey mold lesions might be driven by an increased susceptibility to oxidative stress [21].

*Zymoseptoria tritici* (teleomorph: *Mycosphaerella graminicola*) is the causal agent of Septoria Tritici Blotch (STB), one of the most important foliar diseases of wheat [22,23]. Similar to other Ascomycetes, *Z. tritici* can sense and respond to light by inducing morphological changes during its mycelial development [24,25,26]. Its light-sensing capacity could be ascribable to the multiple photoreceptor homolog genes coded in the *Z. tritici* genome such as the blue-light receptor white collar-1 (*Ztwco-1*), VIVID, cryptochromes, photolyases, and phytochromes [27,28]. In particular, Ztwco-1 regulates light detection and controls the hyphal morphology, production of pycnidia, and micropycnidiospores [25,29]. Similar to other fungi, Ztwco-1 interacts with the homolog of the white collar-2, Ztwco2, to form the white-collar complex (WCC) [25,29]. In *Z. tritici*, the WCC controls the light-dependent expression of several genes such as those involved in the switching from spores to hyphal growth [29]. In addition, the WCC plays a crucial role in pathogenicity, as it integrates external clues of light, temperature, and leaf surface features to initiate host infection [29]. Interestingly, the deletion of *Ztwco-1* delayed infection in wheat, demonstrating the importance of light detection on the virulence of *Z. tritici* [25].

*Z. tritici* not only responds to white light but also to each individual wavelength within the spectrum of visible light. A transcriptome analysis revealed that a third part of the whole *Z. tritici* genome is differentially expressed in response to white, blue (400–530 nm), red (600–700 nm), and dark conditions [28]. Although 50% of these differentially expressed genes do not have any functional annotation, others are related to light perception, pathogenicity, and fungal development. Of particular interest are the expressions of fungal effectors (*Avr3D1*, *AvrStb6*, *Mg1LysM*, and *Mg3LysM*), mitogen-activated protein kinase (MAPK) genes (*MgSlt2* and *MgHog1*), and small-secreted proteins (SSP), which can be up- and downregulated in response to different light conditions and darkness [28].

The high number of light-induced expressed genes related to fungal virulence suggests that light could have a crucial role in the *Z. tritici*–wheat interaction. Therefore, the aim of this study was to elucidate the effect of different light wavelengths on (1) the morphological characteristics (such as pigmentation) and mycelial growth rate of *Z. tritici*, and (2) the pathogen colonization process in bread wheat leaves. To the best of our knowledge, this is a precursory study on the influence of different light wavelengths on *Z. tritici* colonization in bread wheat.

## 2. Materials and Methods

### 2.1. Fungal Strain and Inoculum Production

The *Z. tritici* strain 60.2, characterized and kindly provided by the Department of Agri-Food Science and Technology, University of Bologna, Italy, was used in this study. The fungus, stored at −80 °C, was preliminarily grown on a potato dextrose agar (PDA, Biolife Italiana, Milan, Italy) for 30 days at 22 °C in the dark to produce mycelium. Mycelium plugs for in vitro experiments were obtained from solid media as described in Section 2.3. Conidia for bread wheat artificial inoculation were obtained by using a liquid yeast–glucose (GY) medium (30 g/L glucose (Sigma-Aldrich, St. Louis, MO, USA), 10 g/L yeast extract (Biolife Italiana, Milan, Italy)) inoculated with a mycelium plug (5 mm diameter) and grown in an orbital shaker (250 rpm) for 14 days at 21 °C under a photoperiod of 10 h. Conidia were collected by centrifugation at 3000 rpm for 15 min at 4 °C; their concentration was calculated by using a hemocytometer, and they were immediately used for artificial inoculations.

### 2.2. Light Treatments

This experiment was performed by using four different light spectra: blue (470 nm), red (627 nm), blue–red (50:50), and white light with a photoperiod of 12 h. A light-emitting diode (LED) lighting system based on the Luxeon Rebel LED light source from Philips Lumileds was used in this experiment [30]. The four light spectra were applied at the same cumulative photon flux density of 200 µmol photons/m^2^/s. For both the in vitro and in planta experiments, the temperature (22 ± 1 °C) and humidity (70% RH) were kept constant inside the light chambers [31].

### 2.3. In Vitro Mycelial Growth Assay under Different Light Wavelengths

To analyze the effect of the light wavelength on the mycelial growth of *Z. tritici*, a mycelium plug (5 mm diameter) from the margin of 20-day-old fungal cultures was placed at the center of each PDA plate. The plates were placed in chambers with the four distinct light wavelengths at a distance of 30 cm from the light source and were incubated for 14 days (22 ± 1 °C, 70% RH, and 12 h light). For the dark condition, the inoculated plates were covered in aluminum foil. Each condition had three biological replicates consisting of one 9 cm Petri dish each. The growth area was calculated with the image-processing software ImageJ (version 1.53t). This experiment was repeated twice following the same experimental procedures.

### 2.4. Plant Material and Inoculation

Bread wheat seeds (cv. A416, a commercial spring cultivar with known susceptibility to STB) were surface sterilized in a water–ethanol (95%; Sigma-Aldrich)–sodium hypochlorite (7%; Carlo Erba Reagents, Milan, Italy) sterile solution (82:10:8% volume) for five minutes and rinsed five times with sterile deionized water. The seeds were placed in 14 cm sterile Petri dishes (Aptaca, Canelli, Asti, Italy) containing sterilized filter paper soaked with 10 mL of sterile water and incubated at 4 °C for five days and at 21 °C for two days in the dark to promote germination. Seedlings were sown in pots (length 20 cm, width 20 cm, height 15 cm) containing peat moss, loam soil, and sand (2:2:1 *v*/*v*/*v*). Plants were grown in a glasshouse with a temperature of 18–20/14–16 °C (day/night) and 12 h light. At the tillering stage (Zadok’s scale 25), the plants were artificially inoculated with 10 mL of a fresh conidia suspension of 10^7^ conidia/mL supplemented with 1% Tween^®^ 20 (Sigma-Aldrich). The inoculation was carried out with a hand sprayer. The conidia suspension was sprayed on the leaves three times. After being sprayed, the leaves were left to dry for 20 min each time. Immediately after the third application, the plants were moved into a growth chamber (22 ± 1 °C, 70% RH, and 12 h light) and were covered in plastic bags for 48 h to keep the moisture level around the leaves high and to promote a successful infection. In total, 15 plants per light treatment were inoculated. The inoculated leaves were marked with a permanent marker at the top to differentiate them from new emerging leaves. Fifteen more plants per treatment were sprayed with deionized sterile water as a control. At 48 h post-inoculation, the plastic bags were removed, and the plants were transferred to the different light treatments.

### 2.5. Disease Evaluation

After inoculation, a daily visual evaluation was performed to record symptom appearance. Because of the extension of the symptomless period, we performed the first disease evaluation at 28 days post-inoculation (dpi). At 28 dpi, the incidence of the disease was calculated by counting the number of chlorotic and necrotic lesions. The disease severity values represent the total area of chlorotic and necrotic lesions calculated with the ImageJ software and are expressed as a percentage of the total foliar area. To assess both incidence and severity, three marked leaves per plant were considered as one biological replicate. In total, three plants were evaluated per treatment. The plants remained for 35 dpi under the different light conditions to observe the progress of the disease.

### 2.6. Z. tritici DNA Quantification

The effect of the light wavelengths on *Z. tritici* colonization was evaluated through fungal DNA quantification by a qPCR analysis at 7, 14, 21, and 28 dpi. Three plants were sampled per light condition at each time point. Three leaves per plant were assessed, pooled, and considered as one biological replicate. At 28 dpi, the leaves previously scored for disease incidence and severity were used. The sampled leaves were frozen, freeze dried with a Heto Powder Dry LL3000 freeze-drier (Thermo Fisher Scientific, Waltham, MA, USA), and ground with a Mixer Mill MM200 (Retsch^®^, Dusseldorf, Germany). DNA extraction was performed according to the method described by Covarelli et al. [32]. Briefly, 600 μL of a hexadecyltrimethylammonium bromide (CTAB) buffer ((CTAB 8 g/L, sarkosyl 10 g/L, sorbitol 25 g/L, ethylenediaminetetraacetic acid tetrasodium salt dihydrate (EDTA) 8 g/L, polyvinylpolypyrrolidone (PVPP) 10 g/L, NaCl 87.66 g/L) (all from Sigma-Aldrich)) preheated at 65 °C and 10 μL of RNAse A (20 mg/mL; Invitrogen, Carlsbad, CA, USA) were added to 50 mg of the foliar tissue in 2 mL sterile tubes (Eppendorf, Hamburg, Germany). The samples were homogenized with a Mixer Mill MM200 for 6 min at a frequency of 25 Hz and were placed at 65 °C into a water bath for 30 min. Then, 112.5 μL of isoamyl alcohol:chloroform (1:24 *v*/*v*) (Sigma-Aldrich) solution and 150.75 μL of potassium acetate (5 mol/L) (Sigma-Aldrich) were added to each sample. After vortexing and incubating at −20 °C for 30 min, the samples were centrifuged at 12,470× *g* for 20 min and the upper phase was resuspended in 500 μL of a solution containing 450 μL of isopropanol and 45 μL of sodium acetate (0.1 mol/L) (Sigma-Aldrich). The samples were kept at 4 °C for 15 min. After centrifugation (12,470× *g* for 10 min), DNA pellets were washed twice with 70% ethanol (Sigma-Aldrich) and resuspended in 40 μL of sterile water (Thermo Fisher Scientific). The total extracted DNA was quantified with a NanoDrop^TM^ UV/Vis (Thermo Fisher Scientific) spectrophotometer.

The dilution series from 0.05 g to 50 ng of *Z. tritici* DNA and from 5 pg to 50 ng of A416 bread wheat DNA, with a serial dilution factor of 10, were produced to set up standard curves. Two technical replicates of each dilution were used in each assay. Standard curves were generated by plotting the logarithmic values of the known DNA quantities versus the corresponding cycle threshold (Ct) value. For each standard curve, from the average Ct of each dilution, the line equation (y = mx + q), R^2^ value, and reaction efficiency (10(−1/m)) were calculated. The limit of detection (LOD) of the fungal biomass was 0.05 pg. DNA concentrations were adjusted to 25 ng to perform the qPCR assays. Species-specific primers, ST-rRNA F (5′-CTGCGTCGGAGTTTACGAGT-3′) and ST-rRNA R (5′-GTGAGGGCTCGTGAACTCC-3′), previously reported by Guo et al. [33], were used for the quantification of *Z. tritici* DNA in planta. The qPCR reaction had a total reaction volume of 12 µL, containing 2.5 µL of total DNA, 6 µL of 2 × SYBR^®^ Select Master Mix for CFX (Applied Biosystem, Foster City, CA, USA), 1.5 µL of 2 µM of each primer, and 0.5 µL of sterile DNase-free water. The PCR conditions consisted of 10 min at 95 °C, 45 cycles at 95 °C for 15 s and 61 °C for 1 min, heating at 95 °C for 10 s, cooling at 60 °C, and finally an increase to 95 °C by 0.5 °C every 5 s with the measurement of fluorescence. qPCR assays were performed in a CFX96 Real-Time System (Bio-Rad). Hor1F (5′-TCTCTGGGTTTGAGGGTGAC-3′) and Hor2R (5′-GGCCCTTGTACCAGTCAAGGT-3′) primers [34] were used for the quantification of wheat DNA following the previously described PCR conditions. The *Z. tritici* biomass was expressed as a ratio of fungal DNA (pg) to wheat foliar DNA (ng).

### 2.7. Statistical Analysis

All the statistical analyses were performed by using R Statistical Software (version 4.0; R Foundation for Statistical Computing, Vienna, Austria) [35,36]. The mean ± standard error of the replicates was calculated, and the treatment means were compared by using an ANOVA. The growth data were analyzed by using a two-way ANOVA by considering the light treatment and experiment replicate as independent variables. The disease incidence, disease severity, and *Z. tritici* quantification data were analyzed with a one-way ANOVA. Tukey’s multiple comparison test was performed to test the pairwise contrast (*p* < 0.05).

## 3. Results

### 3.1. Effect of the Different Light Conditions on the Morphology and Growth of Z. tritici Mycelium

Significant differences in the mycelial growth of *Z. tritici* were observed in response to different light wavelengths (Figure 1A). At 14 dpi, the blue light significantly reduced *Z. tritici* growth by 40%, 31%, 30%, and 28% compared to the dark, red, blue–red, and white light, respectively. The colony growth was significantly higher in the dark condition compared to all the light treatments except for the red wavelength condition, whereby similar growth rates were induced. No significant differences were observed under the red, blue–red, and white light. Interestingly, the light conditions tested in this experiment induced morphological differences in *Z. tritici* colonies (Figure 1B). In general, all the colonies produced thick white mycelia with undulate borders and varied aerial development. In particular, the blue wavelength induced the production of abundant aerial mycelium, while the red light led to the formation of concentric rings on the marginal area of the colony. An additive effect was observed under the blue–red light, with the production of aerial and compact mycelia in the center and border of the colonies, respectively. No peculiar differences in pigmentation were observed.

Thus, variations in the colony growth and phenotype provided evidence of *Z. tritici* 60.2 strain responses to light.

### 3.2. Effect of Light on Z. tritici Colonization of Bread Wheat Leaves

The light conditions greatly influenced foliar infection by *Z. tritici* (Figure 2). The first chlorotic lesions were observed at 19 dpi in the inoculated plants grown under white light. At 28 dpi, significant (*p* < 0.05) differences in disease incidence and severity were observed in the plants grown under all light conditions. In terms of incidence, a high number of foliar lesions was observed in the plants grown under white light, followed by those developed under blue and blue–red wavelengths. Minimal or no lesions were detected in the plants grown under red light (Figure 2A). In addition, white light significantly increased the disease severity by 28.7% in the inoculated leaves, in comparison with all the light conditions (*p* < 0.05). In contrast, the red light significantly reduced the severity to 0.83% (*p* < 0.05). No differences were observed between the blue (4.5%) and blue–red (5.5%) treatments (Figure 2B).

In the plants grown under white light, the necrotic lesions exhibited a pronounced chlorotic halo following the coalescence along the leaves (Figure 2C). Under blue light, the lesions showed a reduced size with minimal chlorosis, while under blue–red light, the lesions were characterized by chlorotic spots (Figure 2C).

Leaf senescence prevented disease evaluation at 35 dpi, particularly of plants exposed to white and blue light. Because of an accelerated progress of the STB lesions, the foliar evaluation of the plants exposed to white light was interrupted. In fact, the successful infection culminated with the production of several pycnidia (Figure 2D). On the other hand, the fast senescence observed under blue light was possibly due to the damaging effect of light, as few chlorotic spots were also observed in non-inoculated plants. Nonetheless, the spots in the control plants did not progress to necrotic lesions.

The quantification of *Z. tritici* DNA by qPCR confirmed the presence of the pathogen exclusively in the inoculated plants at all time points.

From 7 to 21 dpi, an increase in the fungal DNA was noticed under the blue and blue–red wavelengths. Nonetheless, during this period, no significant difference was observed among the light treatments at different time points or within a single time point. Still referring to this time period (7–21 dpi), no increment in *Z. tritici* DNA was detected under red light while a discontinuous DNA accumulation was evident under white light (Figure 3). At 28 dpi, the highest fungal DNA amounts were detected under white light, followed by blue and blue–red light, with minimal concentrations under red light. Contrary to what was observed at previous time points, at 28 dpi, a significant difference was observed between the white and red light (*p* < 0.05).

At 28 dpi, the DNA concentrations resembled the disease scoring, and a high correlation level (R = 0.81) between fungal DNA and disease severity was observed (Figure 4).

Overall, a reduction in *Z. tritici* amounts was registered from 21 to 28 dpi in all the light treatments. Particularly, the progression of the *Z. tritici* infection under white light showed a stable increase during the first 14 dpi and a decrease at 21 dpi, before it rose again at 28 dpi. No significant differences in disease development between the blue and blue–red treatments were found, while the lowest *Z. tritici* amounts were detected in the plants treated with red light at all time points.

## 4. Discussion

The results of the present study provide additional evidence of the role of light in the mycelial growth of *Z. tritici* and demonstrate the influence of the light spectrum on its foliar infection in bread wheat. These findings suggest that *Z. tritici* does not only sense light [25,28] but it also responds differently depending on the light wavelengths, both in vitro and in planta.

Previous studies have demonstrated the effect of light and dark conditions on the development and melanization of the aerial hyphae of *Z. tritici* [25,26]. We observed that not only white light but also some other wavelengths could induce morphologic changes in the mycelial growth of *Z. tritici*. Of particular interest was the stressing effect exerted by blue light on the *Z. tritici* strain 60.2 that significantly reduced colony growth and promoted aerial mycelium development. Previous studies reported a variable morphological response in the *Z. tritici* strain IPO323 after exposure to blue light, which stimulated aerial mycelium production on water agar in comparison with red light and darkness [24]. However, no morphological changes induced by different light conditions were observed when evaluating the same strain on PDA [28]. It is well known that the light response is significantly influenced by growth conditions such as the culture media, temperature, and pH. For example, *Z. tritici* IPO323 exhibited a contrasting mycelial growth and hyphal pigmentation when it was grown in PDA and YPDA media (yeast extract, peptone, glucose, and agar) under the same light regimen [25]. Similarly, the *Monilinia fructicola* photoresponse varied as a function of the growth media, as intense illumination reduced the growth rate on the PDA but not in the PDA supplemented with tomato pulp [18].

Even if the photoresponse is highly variable among species, in general, blue light has been recognized for its unfavorable effect on in vitro fungal development [37,38,39]. For example, in *Fusarium solani,* blue light inhibited colony growth by reducing both hyphal branches and length [40]. A similar detrimental effect was observed on *Penicillium digitatum* and *P. italicum* growth [41], whilst in *B. cinerea* and *Aspergillus ficuum*, blue light suppressed conidiation and promoted aerial hyphae formation, leading to sterile fluffy colonies [20,42]. The mechanisms by which blue light may have a fungistatic effect are not well understood. However, some studies proposed that light wavelengths between 320 and 450 nm generate free radicals through photosensitization, and those free radicals might be responsible for cellular damage [43,44].

In this study, light had a significant effect on STB caused by *Z. tritici* either by increasing or diminishing the disease incidence and severity. Whilst white and blue lights favored colonization and the first symptoms were evident at 19 dpi, the blue–red light extended the asymptomatic phase at 23 dpi, and the red light repressed infection.

Interestingly, the symptoms caused by the blue–red treatment strictly remained as chlorotic spots without progressing to the necrotic lesions at least until 35 dpi. This suggests that red light, at 50% of prevalence, is enough to extend the latent period even in the presence of other favorable wavelengths. The latent period, defined as the period between infection and the production of new inoculum [45], depends on several factors associated with the host (e.g., genotype, stage, and tissue age), pathogen (e.g., lifestyle, infection mode, and virulence), as well as external conditions (humidity, temperature, and light) [46,47,48]. In this case, the light spectrum was shown to have a crucial role even if host susceptibility, inoculum concentration, and temperature were favorable to infection. Considering that the latent period directly determines the number of life cycles of a pathogen, our results suggest that certain light wavelengths might have the advantage of controlling *Z. tritici* by interrupting the production of conidia required for secondary infections [49].

Although the *Z. tritici* DNA levels showed a discontinuous increase over time, significant differences were observed at 28 dpi. Likewise, the highest fungal amounts in two of the four analyzed time points were observed under white light, which suggests that visible light (400–700 nm) is particularly required for *Z. tritici* infection. This might also explain why *Z. tritici* pycnidia were produced exclusively under white light.

The sampling of different inoculated plants at each time point could account for the disrupted increasing trend as the disease developed at different rates in different plants. The asynchronous development of *Z. tritici* has been previously reported in which spores continuously germinate, grow, and penetrate or remain ungerminated on the host surface for more than ten days even under favorable conditions [50,51]. For this reason, it could be possible to observe a discontinuous trend in fungal biomass in individual plants subject to the same environmental conditions.

Interestingly, the red light inhibited STB progression, which limited *Z. tritici* colonization in inoculated bread wheat leaves. Since the red light did not affect the in vitro growth of strain 60.2, it can be hypothesized that long wavelengths triggered resistance against *Z. tritici* in wheat plants. A similar induction of resistance was observed after exposure to red light in bean and tomato against *B. cinerea* [13,52,53]; in cucumber against *Sphaerotheca fuliginea* [54] and *Corynespora cassiicola* [55]; in broad bean against *Alternaria tenuissima* [56]; and in rice against *Magnaporthe oryzae* [57], *M. grisea* [58], and *Bipolaris oryzae* [59].

The activation of reactive oxygen species (ROS) metabolism, H_2_O_2_ accumulation, production of antimicrobial compounds (e.g., phytoalexins, flavonoids, anthocyanins), and expression of elements related to both the salicylic acid (SA) and jasmonate acid (JA) signaling pathways are some of the mechanisms involved in the protective effect of red light [60]. Additionally, some studies have reported a crucial role of red/far-red light receptors, the phytochromes, in light-dependent immunity against bacterial and fungal pathogens [61,62,63,64]. For example, in *Arabidopsis thaliana*, both the hypersensitivity response (HR) and systemic acquired response (SAR) require the activity of the phytochromes phyA and phyB in response to *Pseudomonas syringae* infection [61,62]. Similarly, the red-light receptors PHYA, PHYB, and PHYC control the expression of genes related to both the JA and SA pathways against *M. oryzae* in rice [64].

Furthermore, gene expression studies on the host–pathogen interaction might help us understand the underlying processes for red-light-induced resistance in wheat. In this context, an analysis of gene expression such as dual transcriptomic studies could allow us to explore the transcript levels of defense-related genes in wheat as well as the virulence genes in *Z. tritici* [65]. Previous research in rice suggests that the underlying mechanisms of red-light resistance induction are linked to the expression of genes involved in SA synthesis, such as the phenylalanine ammonia-lyase (PAL) [57,66]. Therefore, the analysis of key genes of the SA and JA signaling pathways as well as of defense-related genes such as pathogenesis-related (PR) proteins could be crucial to elucidate the wheat response. On the other hand, the gene expression of levels of virulence factors such MAP kinases (*MgHog1*, *MgSTE11*, *MgSTE7*) [67,68], transcriptional factors (*MgSTE12*, *ZtWor1*) [69], and fungal effectors (*Mg3LysM*, *Mg1LysM*) [70] will clarify how light influences the successful host colonization by *Z. tritici*. Of particular interest is the expression of photoreceptor genes, such as *Ztwco-1,* which not only control fungal development but also the virulence in *Z. tritici* [25,29]. Thus, additional studies on the effect of the light spectrum and the photoreceptors involved would lead to a better understanding of the function of light in a pathogen’s life cycle.

## 5. Conclusions

Our results show that *Z. tritici* senses and responds to different light wavelengths, which modulates the in vitro fungal growth and infection rate in bread wheat leaves. Blue light had a detrimental effect on the *Z. tritici* strain 60.2 by reducing colony growth and altering the mycelial morphology. This study confirms the protective effect of red light against fungal pathogens and gives the first evidence of the detrimental influence of red light on *Z. tritici* foliar infection. Further research is necessary to elucidate the molecular mechanisms underlying the possible red-light-induced resistance against *Z. tritici* in wheat.

## Figures and Tables

**Figure 1 jof-09-00670-f001:**
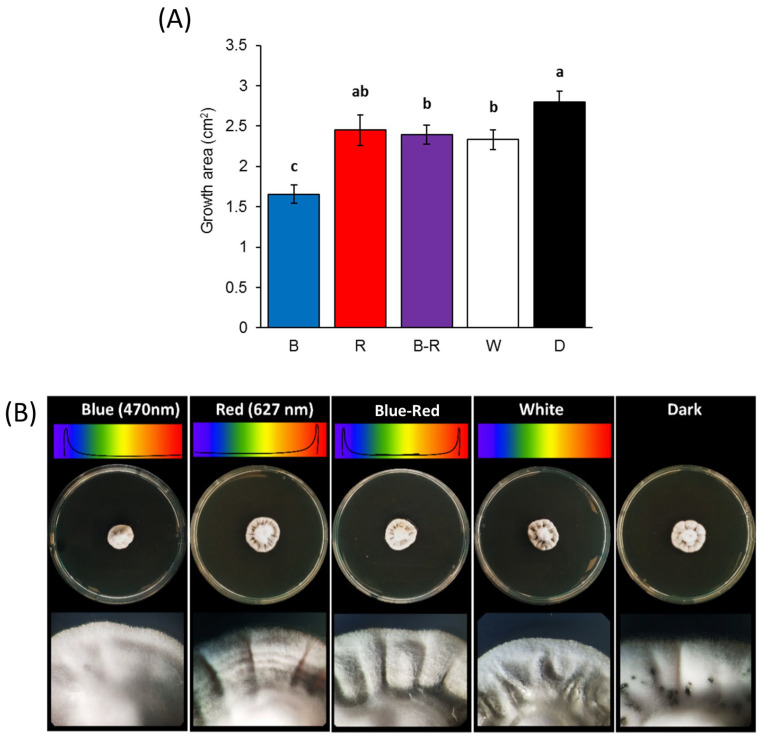
Effect of different light conditions on the mycelial growth of *Zymoseptoria tritici* strain 60.2. A mycelial plug (5 mm diameter) from the margin of a 20-day-old culture was placed at the center of each PDA plate and incubated at 22 °C with 12 h of exposure to blue (B), red (R), blue–red (B-R), or white (W) light or in the dark (D). (**A**) The growth area and (**B**) morphological characteristics were evaluated at 14 dpi. The mean of the growth area of two independent experiments is shown in this figure. The same letters show no significant difference at *p* < 0.05 according to Tukey’s test. Vertical bars indicate the mean ± standard error of the growth area (cm^2^).

**Figure 2 jof-09-00670-f002:**
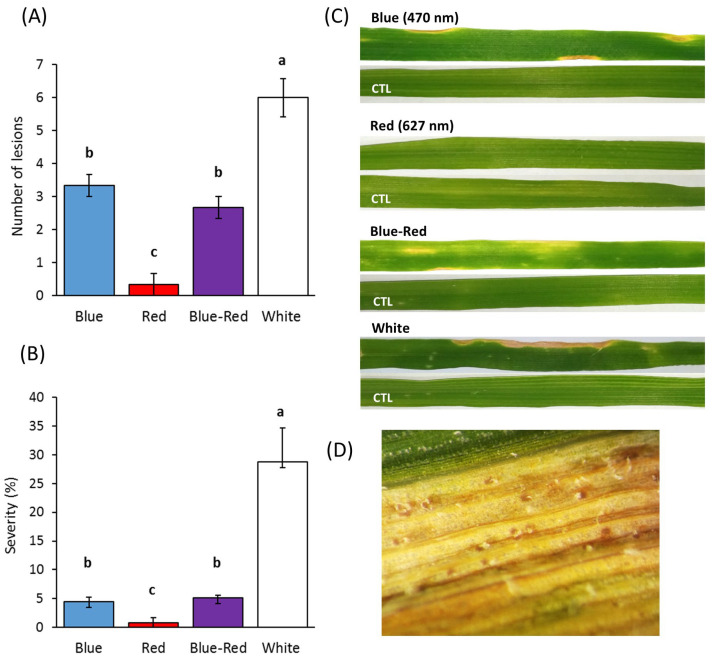
Effect of light wavelengths on Septoria Tritici Blotch in bread wheat plants. Plants at tillering stage (GS25) were artificially inoculated with a 10^7^ conidia/mL *Z. tritici* conidial suspension and exposed to blue, red, blue–red, and white light (200 µmol photons/m^2^/s) under controlled conditions (22 ± 1 °C, 70% RH). Disease incidence (**A**) and severity (**B**) were evaluated at 28 dpi. The same letters show no significant difference at *p* < 0.05 according to Tukey’s test. Vertical bars indicate the mean ± standard error of the replicates. (**C**) STB symptoms at 28 dpi in inoculated and control (CTL) wheat plants under different light conditions. (**D**) Pycnidia produced under white light at 35 dpi.

**Figure 3 jof-09-00670-f003:**
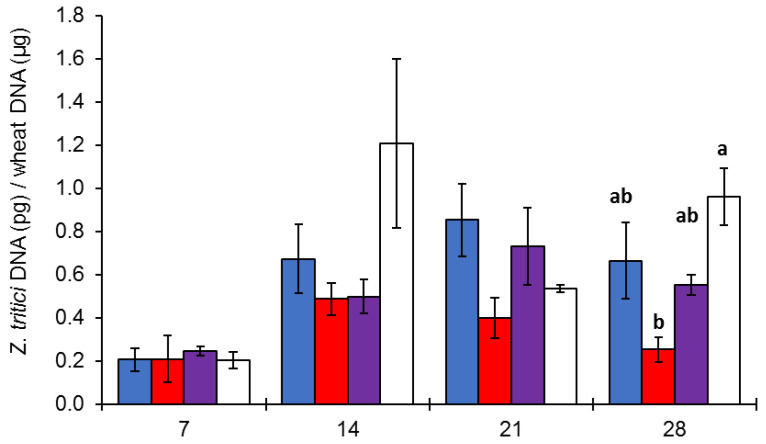
*Z. tritici* progression in inoculated bread wheat plants assessed by qPCR after 7, 14, 21, and 28 dpi. Plants at tillering stage (GS25) were artificially inoculated with a 10^7^ conidia/mL *Z. tritici* conidial suspension and exposed to blue, red, blue–red, and white light (200 µmol photons/m^2^/s) under controlled conditions (22 ± 1 °C, 70% RH). The same letters show no significant difference at *p* < 0.05 according to Tukey’s test. Vertical bars indicate the mean ± standard error of the replicates.

**Figure 4 jof-09-00670-f004:**
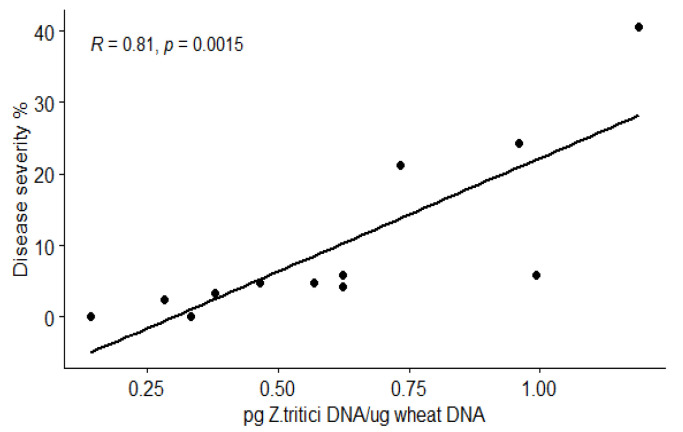
Correlation between disease severity (visually scored at 28 dpi) and *Z. tritici* DNA in bread wheat leaves as detected by qPCR at 28 dpi.

## Data Availability

Data sharing is not applicable to this article.

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
