# Peer review of "Effect of Different Light Wavelengths on Zymoseptoria tritici Development and Leaf Colonization in Bread Wheat"

_jof, 2023, doi:10.3390/jof9060670_

Round 1

Reviewer 1 Report

Title: The title represents the theme of the study;

Abstract: The abstract represents the theme of the study;

Introduction: The introduction managed to explain the research hypothesis.

Material and Methods:

Line 91: gL-1 to g L-1;

Line 98: 470nm to 470 nm;

Line 117: 14-cm? or 14 cm;

Line 152: 10g to 10 g; 8g to 8 g;

Line 157: 30min to 30 min;

Line 161: 450μL to 450 μL;

Line 167: 0.05g to 0.05 g;

Lines 189-195: Insert citation of the R software, package used.

Results: The results obtained are promising for the scientific community and allow a new perspective of reducing the evolution of wheat foliar diseases, with light treatment.

Discussion: Discussion of the results was sufficient to explain the results obtained and allows readers to understand the results obtained.

Conclusion: The conclusion responds to the title and objective of the research, managing to make clear the main results obtained.

Reviewer 2 Report

I also think the work is inspiring and of high quality.

My contributions and criticisms on the article are given in the attached text.

Strain; 1) the virulent level is not clearly reported, 2) it is not possible to "confirm" this research and "check the hypothesis" by different researchers, as it is not known on which resistance genes are virulent or avirulent. This is a "sufficient reason for rejection" of the article and should be evaluated by the editor and authors. The Z. tritici strain 60.2, characterized (line 85), but "no characteristic features are given" or "no reference is made to a literature" has been found.

It is obligatory to give literature regarding the susceptible of cv. A416. As a sustainable control; -Longbow (UK) (Brown JKM, Kema GHJ, Forrer HR, Verstappen ECP, Arraiano LS, Brading PA, Foster EM, Fried PM, Jenny E, 2001. Resistance of wheat cultivars and breeding lines to septoria tritici blotch caused by isolates of Mycosphaerella graminicola in field trials. Plant Pathology 50, 325– 38.)- Riband (UK),

Is there an explanation in the publication on how to calculate disease severity and intensity?

2.1. Fungal Strain and Inoculum Production (line 85-96) section. No literature information was found in this section.

2.3. In vitro Mycelial Growth Assay under Different Light Wavelengths line (105-112) section. No literature information was found in this section.

2.4. Plant Material and Inoculation (114-133) section. No literature information was found in this section.

mL??? (change?) mL−1!!!!!!!! g L???  g/L!!!!

Author Response

We are very grateful to reviewer 2 for her/his detailed revision and comments, which certainly improved the scientific quality of the paper. We have taken into consideration all the suggestions. You can find all corrections in the last version of the manuscript. Due to the high number of comments, we directly replied in the PDF file where the reviewer highlighted her/his comments.

Reviewer 3 Report

The study provides valuable insights into the impact of different light wavelengths on the development and colonization of Zymoseptoria tritici (Z. tritici) in bread wheat. The researchers conducted in vitro and in planta experiments to evaluate the effects of blue, red, blue:red, and white light on fungal growth and morphology. The results indicate that different light wavelengths induce specific morphological changes in Z. tritici, with blue light inhibiting colony growth, while dark and red light favoring fungal development. Additionally, the study demonstrates that light quality influences host plant colonization, with white light stimulating colonization and red light repressing it. The findings contribute to our understanding of the role of light in Z. tritici colonization.

Strengths:

  1. The study includes both in vitro and in planta experiments, providing a more comprehensive understanding of the effects of light on Z. tritici development and colonization in bread wheat.
  2. investigation of morphological changes induced by different light wavelengths enhances our knowledge of how Z. tritici responds to light and sheds light on potential mechanisms underlying the observed effects.
  3. The study's exploration of the influence of light quality on host colonization expands our understanding of the Z. tritici-wheat interaction and provides valuable insights into the dynamics of fungal pathogenesis.
  4. The study contributes new insights by demonstrating the detrimental effect of blue light on Z. tritici growth and the protective effect of red light against foliar infection. These findings have important implications for the management of Z. tritici in wheat crops.

Suggestions for Improvement:

  1. It would be advantageous if the study provided more specific details regarding the proposed directions for future investigations in this area. For instance, exploring the gene expression patterns of Z. tritici under red light conditions and conducting functional analyses of genes associated with pathogenicity could provide insights into the underlying molecular mechanisms driving the observed influence of red light on Z. tritici infection. Additionally, investigating the potential involvement of light-responsive signaling pathways or photoreceptors in mediating the fungal response to red light would contribute to a deeper understanding of the molecular mechanisms at play. Although Z. tritici DNA levels exhibited a discontinuous increase over time, significant differences were observed at 28 days post-inoculation (dpi). Notably, the highest fungal quantities at two out of the four analyzed time points were observed under white light, suggesting that visible light in the range of 400 to 700 nm is particularly essential for Z. tritici infection. This finding may also provide a rationale for the exclusive production of Z. tritici pycnidia under white light conditions, indicating a potential correlation between light quality and fungal reproductive structures. Further investigations into the specific wavelengths and photoreceptors involved in the regulatory mechanisms underlying Z. tritici infection and pycnidia formation would contribute to a more comprehensive understanding of the role of light in the life cycle of this pathogen.
  2. Sample Size and Replication: It would be helpful to provide information about the sample size and replication used in the experiments in the abstract. This would enhance the robustness and reliability of the findings.
  3. Statistical Analysis: The abstract does not mention the statistical analyses employed to support the reported results. It is essential to include information regarding the statistical methods used and the significance levels to ensure the validity of the conclusions drawn.

Comparing this study to that of McCorison and Goodwin (2020) (McCorison, C.B., Goodwin, S.B. The wheat pathogen Zymoseptoria tritici senses and responds to different wavelengths of light. BMC Genomics 21, 513 (2020). https://doi.org/10.1186/s12864-020-06899-y), it is evident that the research focus, namely the effect of light on Zymoseptoria tritici, is comparable. Nonetheless, this new submission adds to and expands upon the findings presented in McCorison and Goodwin (2020). The following are the distinctions and additional insights offered by this new study:

  1. In Vitro and In Planta Analysis: This study includes the analysis of both in vitro and in planta development of Z. tritici under different light conditions. This expands the scope of the study beyond the transcriptomic analysis conducted in McCorison and Goodwin (2020).
  2. Morphological Changes: This study investigates the morphological changes induced by different light wavelengths on Z. tritici growth. It reports that blue light inhibits colony growth, while dark and red light promote fungal development. This provides important insights into the specific effects of different light wavelengths on the morphology of Z. tritici.
  3. Host Colonization: This study examines the influence of light quality on host colonization by Z. tritici. It demonstrates that white light stimulates host colonization, while red light represses it. This aspect further expands our understanding of the interaction between Z. tritici and the host plant.

Considering these distinctions and the additional insights provided by this study, as a reviewer, it would be appropriate to acknowledge the novelty and value of the new study. The research presented in this study builds upon the transcriptomic analysis in McCorison and Goodwin (2020) by incorporating in vitro and in planta experiments, investigating morphological changes, and exploring the effects on host colonization.

Overall Recommendation: Based on the presented study, the study provides valuable insights into the influence of different light wavelengths on Z. tritici colonization in bread wheat. The research approach, inclusion of in vitro and in planta experiments, and the novel findings make it a promising study. However, some minör improvements are necessary, such as providing more details on proposed mechanistic investigations, specifying sample size and replication, and including information on statistical analyses. With these modifications, the study would be a valuable contribution to the field. Additionally, blue-red should be used throughout the text rather than blue:red.

Round 2

Reviewer 2 Report

The entitled manuscript "Effect of Different Light Wavelengths on Zymoseptoria tritici Development and Leaf Colonization in Bread Wheat" was edited by the authors according to the suggestions. The current form can acceptable.